# A New Wearable System for Sensing Outdoor Environmental Conditions for Monitoring Hyper-Microclimate

**DOI:** 10.3390/s22020502

**Published:** 2022-01-10

**Authors:** Roberta Jacoby Cureau, Ilaria Pigliautile, Anna Laura Pisello

**Affiliations:** 1CIRIAF, Interuniversity Research Center on Pollution and Environment Mauro Felli, University of Perugia, 06125 Perugia, Italy; roberta.jacobycureau@studenti.unipg.it (R.J.C.); ilaria.pigliautile@unipg.it (I.P.); 2Department of Engineering, University of Perugia, 06125 Perugia, Italy

**Keywords:** outdoor environmental monitoring, wearable sensing technique, urban microclimate, urban air quality, urban heat island, urban resilience, community resilience

## Abstract

The rapid urbanization process brings consequences to urban environments, such poor air quality and the urban heat island issues. Due to these effects, environmental monitoring is gaining attention with the aim of identifying local risks and improving cities’ liveability and resilience. However, these environments are very heterogeneous, and high-spatial-resolution data are needed to identify the intra-urban variations of physical parameters. Recently, wearable sensing techniques have been used to perform microscale monitoring, but they usually focus on one environmental physics domain. This paper presents a new wearable system developed to monitor key multidomain parameters related to the air quality, thermal, and visual domains, on a hyperlocal scale from a pedestrian’s perspective. The system consisted of a set of sensors connected to a control unit settled on a backpack and could be connected via Wi-Fi to any portable equipment. The device was prototyped to guarantee the easy sensors maintenance, and a user-friendly dashboard facilitated a real-time monitoring overview. Several tests were conducted to confirm the reliability of the sensors. The new device will allow comprehensive environmental monitoring and multidomain comfort investigations to be carried out, which can support urban planners to face the negative effects of urbanization and to crowd data sourcing in smart cities.

## 1. Introduction

The urbanization process is continuously increasing, and it is expected that urban inhabitants will represent 68% of the world’s population by 2050 [1]. At the same time that people are relocating to dense urban areas, these places are becoming uncomfortable cities, which increases health and social risks associated with a lower environmental quality [2].

Built environments in cities are typically different compared to rural surroundings, mainly because of surfaces physical properties. The major consequence is a higher air temperature in urban environments, and this phenomenon is called the urban heat island (UHI) [3]. Urban overheating has serious impacts on urban environmental quality and has been already associated with an increase in the local vulnerability and the heat-related mortality and morbidity [4]. Air quality is another concern in urban areas due to the pollution related to anthropogenic activities (e.g., industrial activities and the transport sector) [5]. Indeed, air pollution is considered the principal environmental risk to human health nowadays [6], and some studies already stated that air pollution exposure is associated with higher levels of mortality and higher incidences of cardiovascular and respiratory diseases [7,8,9,10].

The rapid growth in urbanization and its negative consequences have brought new challenges to urban planners to maintain high living features in these environments [11], which has increased the attention on urban spaces quality [12]. People are exposed to thermal, visual, acoustic, and air quality stimuli simultaneously, but their effects on humans are generally examined separately [13]. Several studies focused on the outdoors investigated the influence of thermal stimuli (e.g., air temperature, relative humidity, wind speed, and solar radiation) on people’s perception in terms of their thermal comfort and sensation [14,15,16,17], and the effects of air pollutants concentrations (e.g., particulate matter, O_3_, NO_x_, and CO) on their air quality perceptions [18,19,20]. Recent experiments conducted in indoor environments have already demonstrated the interactions and crossed effects among different comfort domains [21,22,23,24,25], and this approach is still lacking in the outdoor studies, which will be addressed in this paper.

Pedestrians’ wellbeing involves all comfort domains; thus, urban environmental monitoring concerning different comfort domains is needed [26]. Moreover, the complex morphology and heterogeneity of urban landscapes leads to great variabilities in physical parameters within the same urban space, and understanding these varieties at small scales (hyperlocal) is crucial to propose appropriate mitigation solutions that improve life quality in these environments [27]. To investigate these intra-urban variations, high-spatial-resolution data are needed [27], and nowadays, the most common methods to get them with the required precision involve remote sensing using satellite data [28], fixed weather stations network [29], mobile transects through dedicated equipped vehicles [30], and, more recently, wearable devices [31].

Remote sensing has been used to investigate the land surface temperature and its relation to land cover, mainly associated with the UHI problem [28,32,33,34]. An advantage of this technique is the large spatial coverage of satellites and the repetitive data acquisition [32]. However, some satellite-based observations lack temporal or spatial granularity [2]. A more common approach to characterizing a city in terms of physical parameters is based on fixed weather stations, which provides long-term monitoring that allows verifying temporal variations in the environment [35,36]. Nevertheless, a large number of weather stations are required to catch the spatial variability of the urban environment, and this is not available many times because of equipment high initial and maintenance costs and local constraints [31]. The required data granularity can be achieved through dedicated vehicles equipped with monitoring systems [30,37,38]. This technique results in shorter monitoring campaigns (different from the other methods) to minimize the errors related to mixing spatial and temporal data variability, and it is limited to roadways and parking areas [27]. Despite this, mobile transects are the most suitable method to collect high-spatial-resolution data to evaluate the urban environment heterogeneity, and they allow identifying specific discomfort or pollution sources [31].

To overcome the limitation of road vehicle monitoring, portable/wearable devices to measure physical parameters from a pedestrian perspective have been proposed recently. A wearable device is a tool with strong sensing, processing, storage, and communication capabilities that can be worn by the user [39]. Its great advantage is the possibility to catch the user’s real exposure and enable outlining the spatial–temporal variation of the environmental parameters [40]. In this field, Pigliautile and Pisello [27,31] developed a wearable system that consists of a miniaturized weather station and a GPS settled upon a bike helmet. It monitors the air temperature, relative humidity, wind speed and direction, global solar radiation, atmospheric pressure, lighting, and CO_2_ concentration and can be used while walking and biking. Chokhachian et al. [41] presented a mobile station that measures the air temperature, relative humidity, wind speed, globe temperature, and global solar radiation, besides a GPS. All sensors are carried in a backpack by a person that walks through a specific path. Nakayoshi et al. [42] created a wearable device that monitors environmental parameters (air temperature, relative humidity, wind speed, and short- and long-wave radiation) and physiological (skin temperature, pulse rate, and body motion). Saoutieff et al. [43] and Deng et al. [44] created wearable devices with the objective to monitor the user’s exposure to air pollutants. Xu et al. [45] used a miniaturized device while walking to map an intra-urban PM_2.5_ concentration distribution. Liu et al. [46] detected pollutant sources and the respective emission rates through several mobile users equipped with sensors. Dam et al. [47] developed a wearable air-quality sensor to investigate people’s exposure to air pollution and collect the geographic position, air temperature, relative humidity, and ozone, carbon monoxide, and particulate matter concentrations. All these portable/wearable devices are suitable for deep analyzing one comfort domain but lack monitoring parameters related to other ones. Future comfort investigations should focus on the interactions between domains, and for this purpose, monitoring systems should be able to measure various environmental parameters related to different comfort domains. Besides, according to Salamone et al. [48], wearable sensing techniques dedicated to environmental monitoring should also consider ergonomic design principles and present user-friendly solutions.

Considering this background, a new wearable monitoring system was developed, capable to monitor key physical parameters in outdoor environments on a hyperlocal scale. This article introduces this new wearable device, which measures the air quality, thermal, and visual factors with a high spatial resolution from a pedestrian standpoint for the first time in this field. The characteristics of the system and its embedded sensors were presented, along with several tests that were conducted to verify the reliability and the potential of such a new sensing technique.

## 2. Theoretical Background

The urban lifestyle is influenced by the quality of outdoor environments, as these spaces accommodate several social, cultural, and commercial activities, which raises the importance of providing attractive and comfortable outdoor places [12]. However, the urbanization process intrinsically brings negative consequences to environmental quality, as the UHI [3] and air pollution [5]. These issues are worsened when combined with the phenomena caused by global warming, such as heat waves (HWs). HWs refer to a period of unusual extreme hot weather persisting for certain consecutive days [49] and are becoming more intense, frequent, and longer due to the continued warmer climate [50], being the main cause of weather-related deaths [51]. Some studies have already reported the synergistic interaction between the UHI and HWs. In detail, HWs intensify the impacts and affect the spatial and temporal variability of the UHI [49,52,53,54]. The UHI also interacts with HWs by extending the duration of hot conditions [55]. Therefore, the combined effects with the UHI make the consequences of HWs even more harmful in urban environments [56].

Boosting UHI effects during HWs impacts thermal comfort and heat-related morbidity (e.g., heat exhaustion and heat stroke) and mortality [4]. Health risks are intensified for more vulnerable groups, such as elderly and very young people, subjects with pre-existing medical conditions, low-income groups, ethnic minorities, and socially isolated individuals [57]. Moreover, during HWs, the concentration of air pollutants rises, which also aggravates health risks [58]. Poor environmental quality demands precise interventions to reduce health risks and improve people wellbeing. However, the heterogeneity of urban environments causes hyperlocal variation on microclimate conditions that should be assessed to recommend personalized and specific measures adequate to each place [59]. This analysis requires high-spatial-resolution data to identify microscale variations in the physical parameters [27].

Remote sensing from satellite images is a common method for environmental monitoring. Despite allowing repetitive data acquisition, satellite images are not continuously obtained because of satellite movements. Hence, this technique guarantees continuous spatial data but lacks data on temporal continuity [60]. For this reason, sometimes this method is combined with in situ measurements at a ground level at weather stations that can be used to validate remote sensing information or enhance the identification of temporal variations of physical parameters [27]. In fact, in situ measurements permit continuous monitoring but commonly do not offer high-spatial-resolution data due to the limited number of weather stations [60]. Besides, weather stations are rarely placed within urban areas, and measurements out of these environments are not representative of the urban core [61].

As stated before, a suitable solution to identify the spatial and temporal variation of physical parameters at the same time is the use of wearables for environmental monitoring. However, the current studies using this technique usually encompass only one environmental physics domain in the monitoring. Furthermore, the success of a wearable device depends on the proper selection of sensors that should be adequate for continuous monitoring, even in the short term. For instance, the temperature and relative humidity are usually measured with microelectromechanical system (MEMS) sensors. MEMS are significantly smaller than traditional hygrometers and have integrated amplification and analogue-to-digital convertor (ADC) circuitry [62]. The wind speed is frequently measured with ultrasonic anemometers, which have the advantages of no mechanical structure, no start-up wind speed limitation, and a wide measurement range regarding mechanical and pitot-tube anemometers [63].

For the continuous monitoring of gas concentrations, electrochemical sensors are the most used models. In the case of NO_2_ and O_3_ detection, electrochemical sensors are small-sized and allow detecting low parts per billion (ppb) concentration levels [64]. Other techniques lead to larger sensors or detect gas concentrations only around the parts per million (ppm) level [64]. CO_2_ sensing is usually performed with electrochemical sensors and non-dispersive infrared (NDIR) sensors. NDIR CO_2_ sensors usually have superior long-term stability and high gas specificity, and they represent the largest part of advanced CO_2_ sensors [65]. PM concentrations are hard to measure, because different techniques can result in divergent results [66]. The standard method to measure PM concentrations is the gravimetric method, that is, sampling PM on filters and weighing them before and after the collection, and it is usually presented in daily values as the PM collection occurs in a period of nominally 24 h [67]. For continuous monitoring, laser scattering processes are one of the most used techniques to measure PM concentrations [68].

## 3. New Wearable Equipment for Monitoring Outdoor Environments

### 3.1. General Overview

The developed system was an innovative wearable device for monitoring physical parameters in outdoor environments on a hyperlocal microclimate scale, from a pedestrian’s perspective. The equipment consisted of a set of sensors joined in a kind of backpack, as shown in Figure 1. The backpack design was chosen, because it is more comfortable for the user since the ergonomic design is one of the main aspects that should be considered in wearable devices [48]. The system counts with a GPS that allowed registering the spatial variability of the parameters and making more granular monitoring, a great advantage in comparison to fixed weather stations. This wearable may also be useful for microclimate monitoring in historical cities and protected and sensitive areas, where only a few streets may be acceded by classical vehicles.

The system measured several environmental parameters, such as air temperature, relative humidity, atmospheric pressure, wind velocity and direction, global solar radiation, illuminance, particulate matter concentrations (PM_1.0_, PM_2.5_, and PM_10_), and CO_2_, O_3_, and NO_2_ concentrations. Even though the sensors are suited for outdoor uses, they are protected from direct sunlight with 3D printed customized boxes (except the wind, solar radiation, and illuminance sensors). These boxes were specifically designed to allow natural ventilation and do not influence the sensors’ measurements. They were easily detachable from the wearable system, which facilitated its maintenance.

All sensors were connected to a control unit that collected, stored and sent data to the cloud via Wi-Fi. The system designed was modular, which meant that sensors can be replaced in the future without changing the control unit platform. The control unit power was supplied by a battery that guaranteed more than two hours of autonomy. When the system was turned on, the user could get connected via Wi-Fi with a smartphone, laptop, or any other portable device that supported this connection. Once connected, the user could see a dashboard with sensors measurements in real time and could start an environmental data-recording session. The system dashboard was thought to be user-friendly to facilitate real-time monitoring. When the system was used in the recording mode, the physical parameters were registered every five seconds, and when the registration stopped, the system generated a .csv file with all the measurements. Figure 2 presents the system architecture, and Figure 3 shows the system dashboard (with the air temperature, relative humidity, and pressure sensor as examples) when the wearable monitoring equipment was connected to a smartphone.

### 3.2. Sensors Details

The sensors were selected to have the desired specifications and power consumption to conduct non-permanent monitoring with sizes and weights appropriate for the primary purpose of the equipment, that is, environmental monitoring from a pedestrian’s perspective. Figure 4 shows the position of each sensor in the wearable system, and Table 1 presents their technical information.

The chosen sensor for the temperature, relative humidity, and atmospheric pressure was a MEMS sensor developed for mobile applications. The wind sensor was an ultrasonic anemometer. Global solar radiation was measured with a traditional pyranometer, as well as a light sensor, which is a regular model suitable for outdoor applications. The CO_2_ sensor used the NDIR principle.

The sensors dedicated to monitoring NO_2_ and O_3_ concentrations were the only ones with specific operational requirements, because the electrochemical sensors to measure NO_2_ and O_3_ concentrations available in the market require a preheating time to operate properly. As the developed monitoring system aimed to perform even short-term measurements, long preheating times were not suitable for it; thus, the sensors chosen were the ones with lower preheating time requirements (two hours), according to the manufacturer’s specifications. A dedicated power supply unit was developed only to preliminarily switch on these sensors (belonging to the same module; number 7 in Figure 4) and to provide energy for this preheating process without compromising the battery autonomy of the whole monitoring system. Another concern regarding electrochemical sensors was the cross-sensitivity among O_3_ and NO_2_. The solution adopted was to combine the two sensors, as NO2-A43F responded only to NO_2_ and OX-A431 measured both gases concentrations.

The only sensor that required postprocessing data was the one that measured PM concentrations. A sensor based on laser scattering processes was chosen. Data should be corrected using the mean daily values quantified through the gravimetric method, ideally, in a place close to the one where the monitoring with the wearable system was performed. The correction was a simple multiplication of the measurement by a factor. The factor was the ratio between the mean PM value detected with the wearable system and the respective daily mean value measured through the gravimetric method.

### 3.3. Methods to Verify the Sensors Reliability

The system reliability was checked through comparisons between its sensors and selected reference systems, i.e., a climate chamber, a factory-calibrated compact all-in-one weather station [2], and some individually calibrated sensors. The air temperature, relative humidity, wind velocity, and CO_2_ concentration were compared to those in a climate chamber. The choice for checking the reliability of these sensors in the climate chamber was made because of the possibility to quickly vary the parameters inside it. Therefore, it would be possible to verify how the sensors responded to these variations. The solar radiation, the pressure, and the GPS were compared with the compact weather station. The illuminance and PM concentrations were compared to individual calibrated sensors. NO_2_ and O_3_ sensors were factory-calibrated, so their reliability was not verified.

These comparisons were performed through simultaneous measurements of the developed equipment and the references. The simultaneous monitoring inside the climate chamber lasted four hours, with exception for verifying the wind velocity. For this parameter, a separate test was performed inside the chamber with a fan positioned at the same distance of the wearable monitoring system and the climate chamber sensor.

With the compact weather station, the monitoring was performed at a fixed point on a rooftop twice, during midday and at sunset (about two hours and a half each), to check the reliability of the solar radiation sensor in both conditions. The GPS was checked in a 45 min route with the wearable system and the compact weather station.

The illuminance from the wearable device was compared to that from a calibrated luxmeter that recorded values every one minute. This test was performed inside a room and lasted about one hour. The light conditions were changed during this period (turning the lamps on and off and opening and closing the window shutter) to check how fast the luxmeter in the wearable responded to these variations.

The PM sensor reliability was verified in comparison to those of the sensors in a fixed weather station. This station measured PM_2.5_ and PM_10_ concentrations continuously through optical methods, like the one used in the wearable system, and daily through the standard gravimetric technique, which allows correcting the measurements from the wearable system. The simultaneous monitoring for checking the PM sensor reliability lasted two weeks. The sensor from the wearable system was allocated inside a perforated box that allowed ventilation but protected it from heavy rainfall, and it was directly connected to an electrical grid for a continuous power supply. As PM_1.0_ was not monitored by this reference, the reliability of this measurement was not verified.

The data registered simultaneously with the new system and the references were compared through graphs, the root-mean-square error (RMSE), and the coefficient of variation (CV). The RMSE was defined as the standard deviation of the differences between the expected and predicted samples [69]; it measured the mean error between the reference systems and the new one, comparing the records made at the same time. It had the same unit as the considered parameter. The RMSE was calculated according to Equation (1):(1)$$\mathrm{RMSE}=\sqrt{\frac{1}{n}\sum_{i=1}^{n}{\left(y_{i}-{\hat{y}}_{i}\right)}^{2}},$$
where *y_i_* is the measurement from the reference system, *ŷ_i_* is the measurement from the developed wearable system, and n is the number of measurements.

The CV provided a measurement of how close the values measured by the developed system were to the data obtained with the reference systems. It was expressed in a percentage [70], so the less the CV, the more reliable the sensor. It was calculated according to Equation (2):(2)$$\mathrm{CV}=\frac{\mathrm{RMSE}}{\bar{y}},$$
where *ȳ* is the mean of the values measured by the reference system.

## 4. Results and Discussions

### 4.1. Georeferenced Monitoring

The GPS reliability was checked by performing a route with the compact weather station and the new system. From the starting point, it walked around a building, then to the endpoint and finally back to the starting point. Figure 5 shows the records from both systems. The arrows indicate the directions of the walk. It was noticed that in the green open area the paths are overlaid, while at the beginning and end of the path, near the buildings, there were some differences between the systems. However, this divergence was not larger than the accuracy of the GPS unit in the wearable device (2.5 m). The greatest differences were identified in the first three minutes of walking. Therefore, it was confirmed that the GPS of the developed wearable system was reliable, but it is important to wait three minutes after turning it on to register the positions more accurately.

### 4.2. Hygrothermal and Air Quality Monitoring

Figure 6 shows the measurements performed inside the climate chamber for four hours. They were performed to check the reliability of the air temperature, relative humidity, and CO_2_ concentration sensors. The shaded areas around the temperature and relative humidity indicate the accuracy of each sensor, according to the manufacturer. These three parameters followed the same curve patterns to what were recorded by the climate chamber, especially the temperature sensors that presented values very close to the reference ones. Even though the measurements by the other sensors did not seem close to those by the reference system, the RMSEs and the CVs among the samples were acceptable (Table 2), proving the reliability of the sensors. During this monitoring, the wearable equipment was maintained in a fixed spot inside the chamber, and for keeping the equipment in a vertical position, sensor 2 for the temperature and the relative humidity was slightly covered by the equipment underside. This should explain why the air temperature and the relative humidity registered by sensor 2 were less close to those by the reference values than those by sensor 1. Despite this, the RMSE and the CV were still acceptable for this sensor.

The reliability of the wind velocity sensor was assessed in the same climate chamber, but in a separate experimental session. A fan was positioned at the same distance as the wearable equipment and the chamber sensor, and a 30 min monitoring was performed. The results indicated an RMSE equal to 0.65 m/s and a CV of 17%, with the values measured by the wearable system being slightly higher. This variation was quite high in comparison to those measured by the others. However, an accurate comparison of an air-velocity sensor was much more challenging due to the high variability of the wind field. Even if it was tried to position the wearable and reference sensors at the same distance from the fan, if one sensor was a few millimeters closer, the measurement could be affected. For this reason, this variation was considered acceptable for this sensor.

Figure 7 shows the comparisons of the solar radiations during midday and at sunset measured by the wearable sensor, which were compared to those of the compact weather station. There was a good relation between the developed wearable system and the reference. The wearable equipment gathered slightly higher values, compared to the reference, during the midday campaign (Figure 7a). However, then RMSEs and the CVs (Table 2) for these cases were very low (CV lower than 10%), showing a good approximation between the sensor and the reference. It was noted that the lower the solar radiation, the more accurate the values registered by the wearable equipment. However, attention should be paid to the lowest values when the sensor wrongly registered values close to zero. It could be related to the sensor sensitivity. Therefore, even if the solar radiation sensor is reliable, it is not appropriate for measurements when this parameter is less than 100 W/m². In fact, these values characterize the beginning of the night, a period of the day when usually solar radiation is not a significant parameter to describe the outdoor environment.

The pressure sensors were also assessed in contrast to the compact weather station and presented the lowest CV values among all the sensors (less than 1%), showing that the pressure measurements with the wearable were very consistent.

The PM sensor was evaluated with a calibrated sensor installed in a fixed weather station. The continuous measurements from the wearable system and the calibrated sensor were both made with the optical method. Then, they were corrected with the daily values measured with the gravimetric technique. The measurement trends of the wearable sensor and the reference system was the same, that is, both sensors responded in the same way to PM concentration variations. However, the CV values were higher than 30% for both PM_2.5_ and PM_10_. The box protecting the sensor from the wearable system could have interfered with the measurements and enlarged these differences. However, other studies have already reported that PM concentration can diverge when registered with different sensors, even if they are all based on the optical principle [71,72], which makes the continuous monitoring of these parameters harder. In detail, the studies stated that sensors’ performances can vary with different PM sources and background concentrations, temperatures, relative humidity values, and rainfalls [71,72]. Then, due to these hindering factors and the fact that the sensor in the wearable system followed the same trend as the one in the fixed weather station, this CV was considered acceptable.

### 4.3. Visual Monitoring

The luxmeter was compared with an isolated calibrated sensor that recorded illuminance values every one minute. Figure 8 presents the results of the monitoring, which lasted around two hours. The sensors’ measurements were very close to each other, and the comparison between them resulted in an RMSE equal to 11.46 lx and a CV of 3%, proving the illuminance sensor in the wearable system is reliable.

This system was developed to fill a gap of devices that monitors physical parameters related to different comfort domains in high-spatial and -temporal resolutions. Even though no tests combining measurements from all sensors were performed, individual assessments confirmed the reliability of all of them. Further developments should provide a complete monitoring campaign evaluating the performance of the whole wearable equipment at the same time.

## 5. Conclusions

This paper presented a new wearable device that monitored several environmental parameters related to the air quality in the thermal and visual domains on a hyperlocal scale. The system performed counts with a GPS that associated a precise geographic position to each measurement, allowing detecting the intra-urban variations of all the factors it measured. This wearable device is also helpful for microclimate monitoring in historical cities and protected and sensitive areas, where monitoring systems based on vehicles cannot accede to some streets. The system recorded the measurements every five second, which permitted evaluating temporal data variation, but its user-friendly dashboard also enabled easy environmental real-time monitoring. Various tests were conducted to confirm the reliability of each sensor, and the system provided consistent measurements among all of them.

For the very first time, a wearable device dedicated to the environmental monitoring of three environmental domains was presented. The monitoring performed with this system will support carrying out multidomain comfort studies in outdoor environments. Moreover, it enables a comprehensive characterization of urban spaces that can be effective in identifying environmental vulnerabilities and risks and, consequently, framing resilience plans, which can help urban planners to improve cities’ livability and to establish strategies to boost the experience of tourist spots, for example. Future developments should focus on comprising the acoustics domain in outdoor environmental monitoring by measuring the sound pressure level and evaluating the sound spectrum.

## Figures and Tables

**Figure 1 sensors-22-00502-f001:**
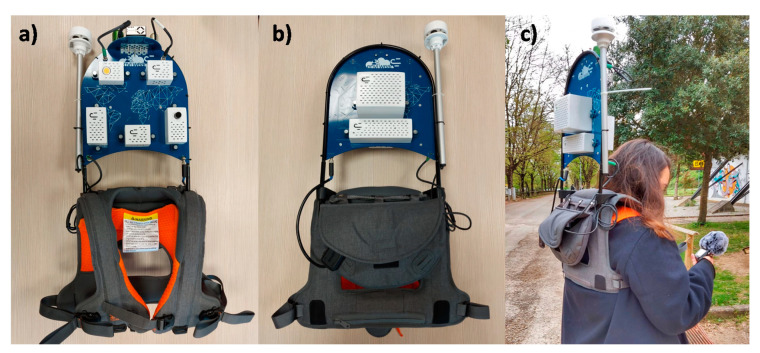
The new wearable system developed for monitoring outdoor environments: (**a**) front view; (**b**) back view; (**c**) system usage during a monitoring campaign.

**Figure 2 sensors-22-00502-f002:**
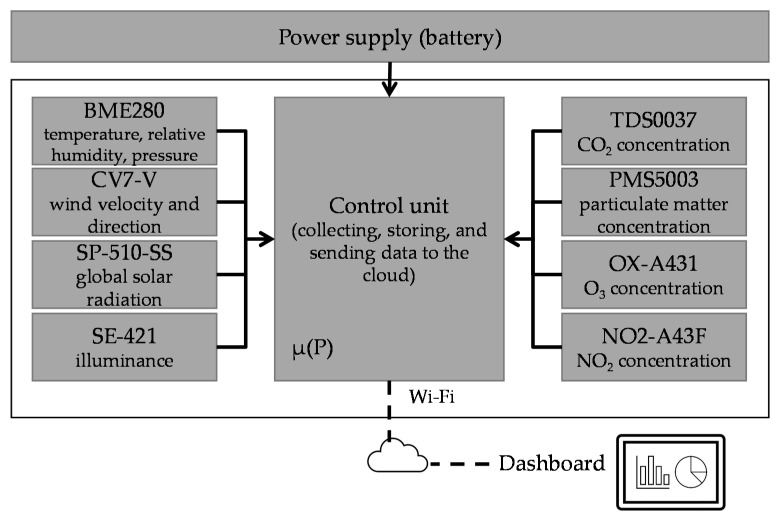
Monitoring system architecture.

**Figure 3 sensors-22-00502-f003:**
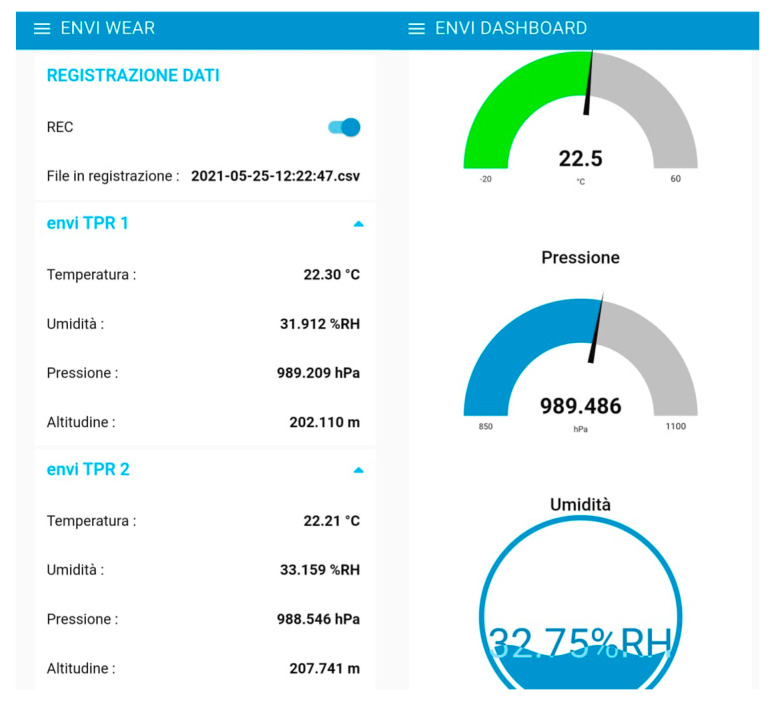
Dashboard to visualize real-time data.

**Figure 4 sensors-22-00502-f004:**
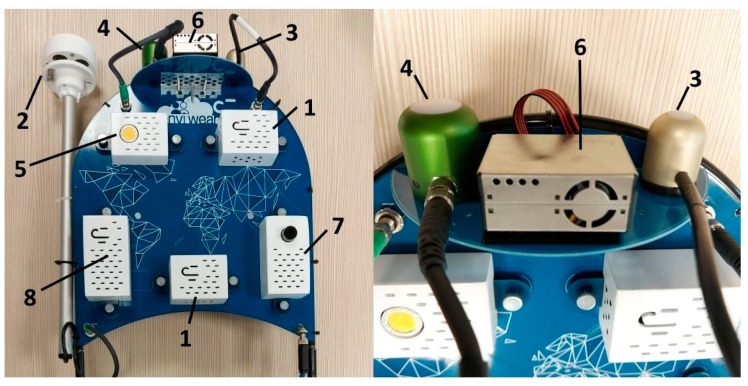
Sensors in the wearable equipment.

**Figure 5 sensors-22-00502-f005:**
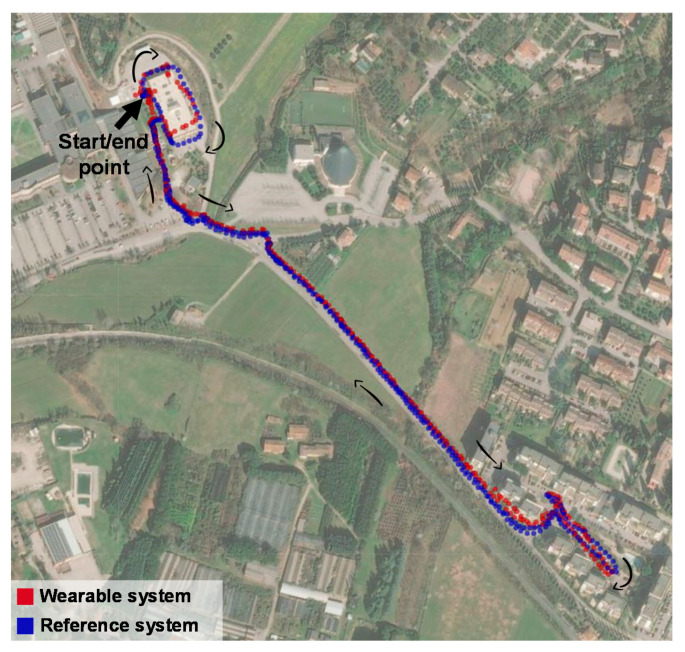
Comparison between the GPS records from the developed wearable and the reference system.

**Figure 6 sensors-22-00502-f006:**
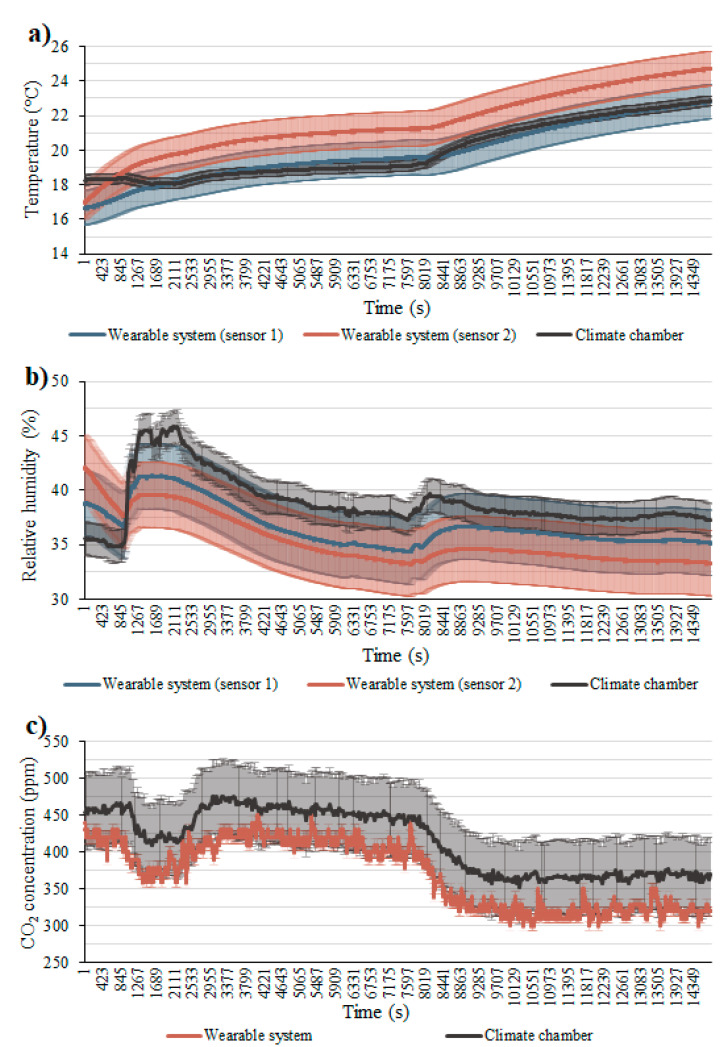
Parameters registered by the developed wearable system and the climate chamber: (**a**) air temperature; (**b**) relative humidity; (**c**) CO_2_ concentration.

**Figure 7 sensors-22-00502-f007:**
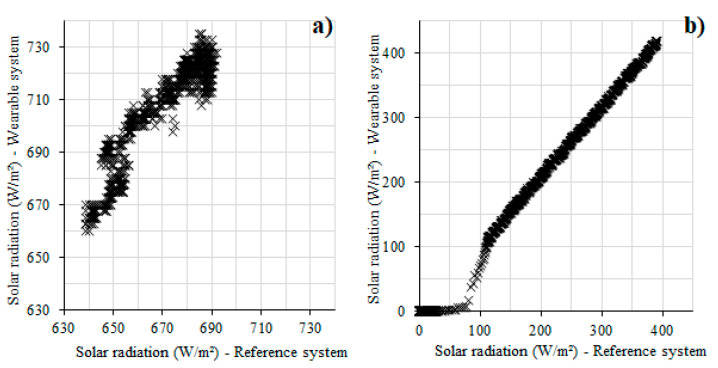
Comparison between the developed wearable system and the reference system (compact weather station) for solar radiation: (**a**) monitoring during midday; (**b**) monitoring at sunset.

**Figure 8 sensors-22-00502-f008:**
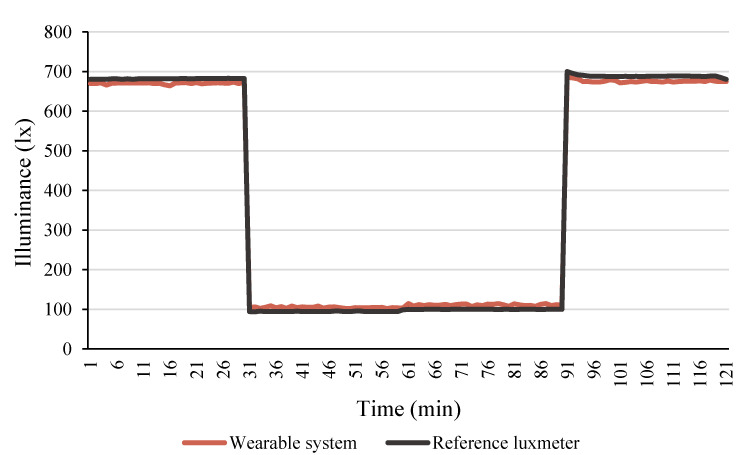
Illuminance registered by the developed wearable system and the reference luxmeter.

**Table 1 sensors-22-00502-t001:** Technical specifications of each sensor in the new wearable monitoring equipment.

ID (Figure 2)	Parameter Monitored	Sensor Model	Technical Specifications
1	Air temperature	BME280	Operation range: −40 °C–85 °C Absolute accuracy: ±1 °C at 0–65 °C
1	Relative humidity (RH)	BME280	Operation range: 10%–90% at 0–65 °C Absolute accuracy: ±3% at 20%–80% RH Response time: 1 s
1	Atmospheric pressure	BME280	Operation range: 300 hPa–1100 hPa at 0–65 °C Sensitivity error: ±0.25%
2	Wind velocity	CV7-V	Operation range: 0.25 Kt–80 Kt Sensitivity: 0.25 Kt Resolution: 0.1 Kt Output update: 2 per second
2	Wind direction	CV7-V	Sensitivity: ±1° Resolution: 1° Output update: 2 per s
3	Global solar radiation	SP-510-SS	Measurement range: 0–2000 W/m² Calibration uncertainty: ±5% Detector response time: 0.5 s Spectral range: 385 nm–2105 nm
4	Illuminance	SE-421	Measurement range: 0–150,000 lx Calibration uncertainty: ±5% Response time: 0.6 s
5	CO_2_ concentration	TDS0037	Accuracy: ±2% at 20 °C Pressure: 1 bar Applied gas: 2.5% volume CO_2_ Response time t_90_: <30 s at 20 °C
6	Particulate matter concentrations (PM_1.0_, PM_2.5_, and PM_10_)	PMS5003	Effective range (PM_2.5_ standard): 0–500 μg/m³ Resolution: 1 μg/m^3^ Maximum consistency error (PM_2.5_ standard): ±10 μg/m³ at 0–100 μg/m³; ±10% at 100–500 μg/m³ Total response time: <10 s
7	O_3_ concentration	OX-A431	Sensitivity (nA/ppm at 1 ppm O_3_): −200 to −650 Response time (t_90_ (s) from zero to 1 ppm O_3_): <80 s
7	NO_2_ concentration	NO2-A43F	Sensitivity (nA/ppm at 2 ppm NO_2_): −175 to −500 Response time (t_90_ (s) from zero to 2 ppm NO_2_): <80 s Range (ppm NO_2_ Limit of Performance Warranty): 20 ppm
8	GPS unit	NEO-M8	Horizontal spatial accuracy: 2.5 m

**Table 2 sensors-22-00502-t002:** Root-mean-square error (RMSE) and coefficient of variation (CV) of the developed wearable system parameters compared to those of the reference systems.

Parameter Monitored	RMSE (lx)	CV
Air temperature by sensor 1 (°C)	0.46	2%
Air temperature by sensor 2 (°C)	1.74	9%
RH by sensor 1 (%)	2.70	7%
RH by sensor 2 (%)	4.20	11%
CO_2_ concentration (ppm)	49.42	12%
Wind velocity (m/s)	0.65	17%
Solar radiation at midday (W/m²)	37.70	6%
Solar radiation at sunset (W/m²)	15.83	11%
Pressure by sensor 1 (hPa)	1.53	0.2%
Pressure by sensor 2 (hPa)	0.76	0.1%
Illuminance (lx)	11.46	3%
PM_2.5_ (µg/m^3^)	2.37	31%
PM_10_ (µg/m^3^)	7.03	38%

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
