# Peer review of "A New Wearable System for Sensing Outdoor Environmental Conditions for Monitoring Hyper-Microclimate"

_sensors, 2022, doi:10.3390/s22020502_

Round 1
Reviewer 1 Report
Dear Authors,
After reviewing your manuscript, I think it can be interesting to the Journal's readers and it can contribute to the existing body of literature. However, minor to moderate revisions are required.
Please see the attached word document for details.
Kind regards,
Reviewer

Author Response
Reviewer #1: Dear Authors,
After reviewing your manuscript, I think it can be interesting to the Journal's readers and it can contribute to the existing body of literature. However, minor to moderate revisions are required.
Please see the attached word document for details.
Kind regards,
Reviewer
Authors:
The Authors thank the Reviewer for the careful reading of the manuscript and for the comment that improved the quality of the manuscript. The comments pointed in the attached document have been addressed by the Authors in the reviewed version of the manuscript (red colour), as specified below.
R1.1:
Title: Adequate and it appropriately describes the manuscript.
Abstract and Keywords: Informative, concise and includes all the relevant information.
Section 1. Introduction: The Introduction section:
- addresses urbanization and urban environment
- notes quality of life, air quality, heat-related human health risks
- presents the use of wearable devices for intra-urban variations and parameters
- addresses remote sensing for land surface temperature
- presents the possibilities of wearable devices
- presents previous studies in this domain
Overall, the Introduction section includes relevant information in a structured manner.
Please note and fix that the title of Section 2. New wearable equipment for monitoring environment (line 117) is accidentally included in the last paragraph (from line 111 to 118). As a result, the last paragraph is in bold text - not formatted properly.
Authors:
The Authors thank the Reviewer for the positive feedback regarding the Title, Abstract and Keywords, and Introduction. The format of the last paragraph of Section 1 and title of Section 2 were fixed (lines 111 to 117).
R1.2:
Section 2. New wearable equipment for monitoring outdoor environment: This section provides a general overview of the wearable system and details on the sensor.
Section 3. Methods: This section and the previous section could be presented as one section. The content in Section 2 and Section 3 are similar in style and presentation.
Authors:
The Authors merged Sections 2 and 3 as suggested by the Reviewer in the new Section 3 of the revised manuscript. Previous section 3 becomes “3.3 Methods to verify sensors reliability”.
R1.3:
I propose that Section 2 is renamed to Theoretical background which will include additional detail on urban lifestyle and environment hazards and theoretical overview on remote sensing and sensors. Section 3 should include everything relevant to the specific wearable devices and sensors that are currently presented in Section 2.
In sum, re-structuring and additional theoretical detail is recommended. Should be shorter, concisely presenting theory, while Section 3 should include figures and tables.
Authors:
The Authors included a Section “2. Theoretical background”, providing a theoretical overview related to heat waves and their impacts on urban environmental quality and sensing techniques to monitor physical parameters. Some theoretical information regarding sensors models that were in Section 3 were moved to this new Section 2.
Below, the whole Section 2 (lines 118 to 179).
“The urban lifestyle is influenced by the outdoor environments quality as these spaces accommodate several social, cultural, and commercial activities, which raises the im-portance of providing attractive and comfortable outdoor places [12]. However, the ur-banization process intrinsically brings negative consequences to environmental quality, as the UHI [3] and air pollution [5]. These issues are worsened when combined with the phenomena caused by global warming, such as heat waves (HWs). HWs refer to a period of unusual extreme hot weather persisting for certain consecutive days [49], and are be-coming more intense, frequent, and longer due to the continued warmer climate [50], be-ing the main cause of weather-related deaths [51]. Some studies have already reported the synergistic interaction between UHI and HWs. In details, HWs intensify the impacts and affect the spatial and temporal variability of UHI [49, 52-54]. The UHI also interacts with HWs by extending the duration of hot conditions [55]. Therefore, the combined effects with the UHI make the consequences of HWs even more harmful in urban environments [56].
Boosting UHI effects during HWs impact thermal comfort and heat-related morbidity (e.g., heat exhaustion and heat stroke) and mortality [4]. Health risks are intensified for the more vulnerable groups, such as elderly and very young people, subjects with pre-existing medical conditions, low-income groups, ethnic minorities, and socially isolated individu-als [57]. Moreover, during HWs, the concentration of air pollutants raises, which also ag-gravates the health risks [58]. Poor environmental quality demands precise interventions to reduce health risks and improve people wellbeing. However, the heterogeneity of urban environments causes hyperlocal variation on microclimate conditions that should be as-sessed to recommend personalized and specific measurements adequate to each place [59]. This analysis requires high-spatial-resolution data to identify micro-scale variations in the physical parameters [27].
Remote sensing from satellite images is a common method for environmental moni-toring. Despite allowing repetitive data acquisition, satellite images are not continuously obtained because of satellite movements. Hence, this technique guarantees continuous spatial data, but lacks on temporal continuity [60]. For this reason, some-times this meth-od is combined with in situ measurements at ground level through weather stations that can be used to validate the remote sensing information or enhance the identification of temporal variations of physical parameters [27]. In fact, in situ measurements permits continuous monitoring, but commonly does not offer high-spatial-resolution data due to the limited number of weather stations [60]. Besides, weather stations are rarely placed within urban areas, and measurements out of these environments are not representative of the urban core [61].
As stated before, a suitable solution to identify spatial and temporal variation of physical parameters at the same time is the use of wearables for environmental monitor-ing. However, the current studies using this technique usually encompass just one envi-ronmental physics in the monitoring. Furthermore, the success of a wearable device de-pends on the proper selection of sensors, that should be adequate for continuous moni-toring, even short-term. For instance, temperature and relative humidity are usually measured with microelectromechanical system sensors (MEMS) are significantly smaller than traditional hygrometers and have integrated amplification and Analogue-to-Digital Convertor (ADC) circuitry [62]. Wind speed is frequently measured with ultrasonic ane-mometers, which have the advantages of no mechanical structure, no start-up wind speed limitation, and a wide measurement range regarding mechanical and pitot-tube ane-mometers [63].
For continuous monitoring of gas concentrations, electrochemical sensors are the most used models. In the case of NO2 and O3 detection, electrochemical sensors are small-size and allow detecting low parts per billion (ppb) concentration levels [64]. Other techniques lead to larger sensors or detect the gas concentration only around the parts per million (ppm) level [64]. CO2 sensing is usually done with electrochemical sensors and non-dispersiveinfrared (NDIR) sensors. NDIR CO2 sensors usually have superior long-term stability and high gas specificity, and they represent the largest part of the ad-vanced CO2 sensors [65]. PM concentration is hard to measure because different tech-niques can result in divergent results [66]. The standard method to measure PM concen-tration is gravimetric method, that is sampling PM on filters and weighing them before and after the collection, and it is usually presented in daily values as the PM collection occurs in a period of nominally 24 h [67]. For continuous monitoring, laser scattering pro-cesses is one of the most used techniques to measure PM concentration [68].”
R1.4:
Section 4. Results and discussion: Provides sufficient detail. However, it is proposed to address potential limitations and advantages of this current study. Additionally, the contribution of this study could be addressed.
If possible, the results section and discussion section could be two separate sections.
Authors:
The results and discussions were not separated in two sections because authors believe that this form of presentation makes reading more fluid and facilitates the comprehension of results related to sensors reliability.
A final paragraph addressing the limitations and contribution of this study was added in the end of Section 4.2 according with this Reviewer suggestion to conclude the presentation of results, as follows (lines 408 to 413):
“This system was developed to fill a gap of devices that monitors physical parameters related to different comfort domains in a high-spatial and -temporal resolution. Even though no tests combining measurements from all sensors were performed, individual assessments confirmed the reliability of every sensors. Further developments should provide a complete monitoring campaign evaluating the performance of the whole wearable equipment at the same time.”
R1.5:
Section 5. Conclusion: Appropriate. Concisely and well structured.
References: Adequate. Includes relevant literature sources.
Authors:
The Authors thank the Reviewer for the positive feedback regarding the Conclusion and the choice of references.
Reviewer 2 Report
The article describes a new integrated device monitoring environmental parameters, showing its application. According to the reviewer:
- The article is well organized and legible.
- It contains essential elements concerning the analyzed problem.
- The description of parameters, measurement, and analysis is sufficient in the current topic.
- The obtained accuracy of the measurements does not raise any objections either, mainly as they refer to more significant areas where the obtained result is sufficient (research facilities are not laboratories).
- The reviewer sees the potential in the described device, especially in the protection, shaping and management of space in a sustainable manner.
- Please check the English spelling.
To sum up:
The article is a very good and helpful study. It seems that further analysis and studies for specific examples will be carried out. The Authors mention historic cities. I also want to highlight protected and sensitive areas. I am eager to read the following studies. Good luck.
Author Response
Reviewer #2: The article describes a new integrated device monitoring environmental parameters, showing its application. According to the reviewer:
- The article is well organized and legible.
- It contains essential elements concerning the analyzed problem.
- The description of parameters, measurement, and analysis is sufficient in the current topic.
- The obtained accuracy of the measurements does not raise any objections either, mainly as they refer to more significant areas where the obtained result is sufficient (research facilities are not laboratories).
- The reviewer sees the potential in the described device, especially in the protection, shaping and management of space in a sustainable manner.
- Please check the English spelling.
To sum up:
The article is a very good and helpful study. It seems that further analysis and studies for specific examples will be carried out. The Authors mention historic cities. I also want to highlight protected and sensitive areas. I am eager to read the following studies. Good luck.
Authors:
The Authors thank the Reviewer for the careful reading of the manuscript and for the positive feedback regarding the paper. The Authors carefully checked the English spelling. The modifications made in the manuscript related to this comment are also in the “Track Changes” mode. Protected and sensitive areas were also highlighted as places where the use of a wearable device may be the appropriate solution for environmental monitoring (lines 190 and 419).